



# 1 Seasonal Variation of Oxygenated Organic Molecules in Urban
# 2 Beijing and their Contribution to Secondary Organic Aerosol

Yishuo Guo[1], Chao Yan[1,2,*], Yuliang Liu[3], Xiaohui Qiao[4], Feixue Zheng[1], Ying Zhang[1], Ying Zhou[1],
Chang Li[1], Xiaolong Fan[1], Zhuohui Lin[1], Zemin Feng[1], Yusheng Zhang[1], Penggang Zheng[5], Linhui Tian[7],
Wei Nie[3], Zhe Wang[5,6], Dandan Huang[8], Kaspar R. Daellenbach[2,9], Lei Yao[1,2], Lubna Dada[2,9], Federico
Bianchi[2], Jingkun Jiang[4], Yongchun Liu[1], Veli-Matti Kerminen[2], Markku Kulmala[1,2]
**Affiliations:**
[1] Aerosol and Haze Laboratory, Beijing Advanced Innovation Center for Soft Matter Science and Engineering, Beijing
University of Chemical Technology, Beijing, China
[2] Institute for Atmospheric and Earth System Research / Physics, Faculty of Science, University of Helsinki, Finland
[3] Joint International Research Laboratory of Atmospheric and Earth System Research, School of Atmospheric Sciences,
Nanjing University, Nanjing, China
[4] State Key Joint Laboratory of Environment Simulation and Pollution Control, State Environmental Protection Key Laboratory
of Sources and Control of Air Pollution Complex, School of Environment, Tsinghua University, Beijing, China
[5] Division of Environment and Sustainability, The Hong Kong University of Science and Technology (HKUST), Hong Kong
SAR, China
[6] Department of Civil and Environmental Engineering, The Hong Kong Polytechnic University (HKPolyU), Hong Kong SAR
[7] Department of Civil and Environmental Engineering, Faculty of Science and Technology, University of Macau, Taipa, Macau,
China
[8] State Environmental Protection Key Laboratory of Formation and Prevention of Urban Air Pollution Complex, Shanghai
Academy of Environmental Sciences, Shanghai, China
[9] Laboratory of Atmospheric Chemistry, Paul Scherrer Institute, Villigen, Switzerland.
*Correspondence to: Chao Yan (chao.yan@helsinki.fi)*
**Abstract** Oxygenated organic molecules (OOMs) are crucial for atmospheric new particle formation and secondary
organic aerosol (SOA) growth. Therefore, understanding their chemical composition, temporal behavior, and
sources is of great importance. Previous studies on OOMs mainly focus on environments where biogenic sources
are predominant, yet studies on sites with dominant anthropogenic emissions, such as megacities, have been lacking.
Here, we conducted long-term measurements of OOMs covering four seasons of the year 2019 in urban Beijing.
The OOM concentration was found to be the highest in summer ($1.6\times10^8$ cm$^{-3}$), followed by autumn ($7.9\times10^7$ cm$^{-3}$
), spring ($5.7\times10^7$ cm$^{-3}$) and winter ($2.3\times10^7$ cm$^{-3}$), suggesting that enhanced photo-oxidation together with the rise
of temperature promote the formation of OOMs. Most OOMs contained 5 to 10 carbon atoms and 3 to 7 effective
oxygen atoms ($nO_{eff}=nO-2\times nN$). The average $nO_{eff}$ increased with increasing atmospheric photo-oxidation capacity,
which was the highest in summer and the lowest in winter and autumn. By performing a newly developed workflow,
OOMs were classified into four types: aromatic OOMs, aliphatic OOMs, isoprene OOMs, and monoterpene OOMs.
Among them, aromatic OOMs (29-41 %) and aliphatic OOMs (26-41 %) were the main contributors in all seasons,
indicating that OOMs in Beijing were dominated by anthropogenic sources. The contribution of isoprene OOMs
increased significantly in summer (33 %), which is much higher than those in other three seasons (8-10 %).



Concentrations of isoprene ($0.2\text{-}5.3\times10^7$ cm$^{-3}$) and monoterpene ($1.1\text{-}8.4\times10^6$ cm$^{-3}$) OOMs in Beijing were lower
than those reported at other sites, and they possessed lower oxygen and higher nitrogen contents due to high NO$_x$
levels (9.5-38.3 ppbv) in Beijing. With regard to the nitrogen content of the two anthropogenic OOMs, aromatic
OOMs were mainly composed of CHO and CHON species, while aliphatic OOMs were dominated by CHON and
CHON$_2$ ones. Such prominent differences suggest varying formation pathways between these two OOMs. By
combining the measurements and an aerosol dynamic model, we estimated that the SOA growth rate through OOM
condensation could reach 0.64 μg·m$^{-3}$·h$^{-1}$, 0.61 μg·m$^{-3}$·h$^{-1}$, 0.41 μg·m$^{-3}$·h$^{-1}$, and 0.30 μg·m$^{-3}$·h$^{-1}$ in autumn, summer,
spring, and winter, respectively. Despite the similar concentrations of aromatic and aliphatic OOMs, the former had
lower volatilities and, therefore, showed higher contributions (46-62%) to SOA than the latter (14-32%). By contrast,
monoterpene OOMs and isoprene OOMs, limited by low abundances or high volatilities, had low contributions of
8-12% and 3-5%, respectively. Overall, our results improve the understanding of the concentration, chemical
composition, seasonal variation and potential atmospheric impacts of OOMs, which can help formulate refined
restriction policy specific to SOA control in urban areas.



## 1. INTRODUCTION

Atmospheric aerosols affect global climate both directly and indirectly (Stocker, 2014) and are known to have a detrimental influence on human health (Lelieveld et al., 2015). Modeling studies have suggested that new particle formation (NPF) dominates the number concentration of particles and is an important contributor to cloud condensation nuclei (CCN) in the global atmosphere (Merikanto et al., 2009;Gordon et al., 2017). In terms of aerosol mass, it has been shown that a significant fraction is composed of secondary organic aerosol (SOA) (Zhang et al., 2007;Jimenez et al., 2009;Hallquist et al., 2009). In both NPF and SOA formation processes, oxygenated organic molecules (OOMs) have been acknowledged as an important contributor, and thus advanced understanding of OOMs is crucial.

The role of OOMs in NPF was first suggested in 2002, but they could not be identified or quantified at that time (O'Dowd et al., 2002). Then, the emergence of atmospheric pressure interface time-of-flight (APi-TOF) mass spectrometer (Junninen et al., 2010;Ehn et al., 2010;Ehn et al., 2012) and Chemical Ionization-APi-TOF (CI-APi-TOF) mass spectrometer (Jokinen et al., 2012;Ehn et al., 2014) provided the first direct measurement of OOMs, which inspired later studies on their role in NPF and SOA growth (Kulmala et al., 2013;Schobesberger et al., 2013;Riccobono et al., 2014;Ehn et al., 2014;Kirkby et al., 2016;Tröstl et al., 2016;Bianchi et al., 2016;Lehtipalo et al., 2018;Rose et al., 2018;Stolzenburg et al., 2018;Mohr et al., 2019;Heinritzi et al., 2020;Yan et al., 2020;Caudillo et al., 2021). These studies found that the functionality and volatility of OOMs are the key factors in determining whether OOM species can participate in NPF (Donahue et al., 2013). More specifically, ultra-low-volatility organic compounds (ULVOCs, whose mass saturation concentrations, $C^*$, are smaller than $3\times10^{-9}$ $\mu g \cdot m^{-3}$, or number saturation concentrations, $N^*$, are smaller than 6 $cm^{-3}$ by assuming an average molar mass of 300 Da) are the main participator of the initial nucleation at certain conditions (Schervish and Donahue, 2020), and extremely-low-volatility organic compounds (ELVOCs, $3\times10^{-9} < C^* < 3\times10^{-4}$ $\mu g \cdot m^{-3}$, $6 < N^* < 6\times10^{5}$ $cm^{-3}$) and low-volatility organic compounds (LVOCs, $3\times10^{-4} < C^* < 0.3$ $\mu g \cdot m^{-3}$, $6\times10^{5} < C^* < 6\times10^{8}$ $cm^{-3}$) can have a dominant contribution to the growth of newly formed particles.

Owing to the significance of OOMs in atmospheric aerosols formation and growth, its reliable measurement is of high importance. Up till now, the majority of reported sites in the lower troposphere with OOM measurement are non-urban areas, such as forest, agricultural pasture and countryside, where the most abundant OOM species are oxidized products from monoterpenes and isoprene. In the boreal forest of southern Finland, the reported OOM concentration was the highest in summer ($4.6\times10^{8}$ $cm^{-3}$) (Huang et al., 2020), followed by autumn ($8.0\times10^{7}$ $cm^{-3}$) (Zha et al., 2018) and spring (~$4.0\times10^{7}$ $cm^{-3}$) (Yan et al., 2016;Roldin et al., 2019;Bianchi et al., 2017). The level of OOMs also varied significantly at different sites. In Melpitz agricultural-forest of central Europe, OOM concentration ($2.5\times10^{8}$ $cm^{-3}$ in summer) (Mutzel et al., 2015) was comparable with that in Hyytiälä, while in Alabama forest of the United States, OOM concentration was much higher ($4.8\times10^{9}$ $cm^{-3}$ in summer) (Massoli et al., 2018;Krechmer et al., 2015), possibly due to higher UVB and temperature. Besides, monoterpene OOMs at





agricultural-rural mixed Vielbrunn were also detected ($3.6 \times 10^6$ cm$^{-3}$ in spring), and results showed that many other
unidentified species also took a large fraction, especially at night (Kürten et al., 2016). All of these studies highlight
the importance of OOM measurement worldwide. Several urban observations were also reported (Brean et al.,
2019;Ye et al., 2020). Although they showed that OOMs in Chinese urban cities contain a significant fraction of
compounds with 6 to 9 carbons and that many contain nitrogen, they either reported concentrations of a few chosen
species or just spectral signals. Moreover, due to the limitation of short measurement periods, they were incapable
of exploring the seasonal behavior of OOM concentration and detailed composition, which are crucial for fully
evaluating their potential contribution to the growth of SOA.
In this work, we studied the OOMs measured by a CI-APi-TOF mass spectrometer using nitrate (NO$_3^-$) as reagent
ions. The dataset covers four seasons of Year 2019. We performed detailed molecular analyses within the mass to
charge ratio between 200-400 Th and identified around 1000 OOMs for each season. The seasonal variations of
their concentration, molecular composition, volatility distribution, and potential SOA contribution were
systematically investigated for the first time. Furthermore, with a newly developed workflow, we traced their
potential sources, including aromatics, aliphatics, monoterpenes, and isoprene. Finally, we evaluated the relative
contribution of anthropogenic and biogenic sources in different seasons.

## 2. MEASUREMENTS AND METHODS

### 2.1. Measurements

The measurement was conducted at the west campus of Beijing University of Chemical Technology (39.95° N,
116.31° E) on the fifth floor of the teaching building, which is about 15 m above the ground level. This station is a
representative urban site, and a detailed description can be found elsewhere (Liu et al., 2020;Yan et al., 2021;Guo
et al., 2021).
The concentration of OOMs was measured by a nitrate (NO$_3^-$)-CI-APi-TOF mass spectrometer (abbreviated as
nitrate CIMS) (Aerodyne Research, Inc.). The basic working principle of this instrument can be found elsewhere
(Jokinen et al., 2012),  and the detailed sampling configuration is the same as that reported by Yan et al. (Yan et al.,
2021). Two steps were included in the quantification of OOM concentration. First, a mass-dependent transmission
experiment was conducted according to a previous study (Heinritzi et al., 2016), and the transmission curve was
obtained by comparing the decrease of primary ion signals and the increase of added perfluorinated acid signals.
Second, the calibration factor of sulfuric acid was applied to estimate OOM concentration. Some studies have shown
that OOM molecules with less oxygen number are not ionized as efficiently as sulfuric acid by NO$_3^-$ (Hyttinen et
al., 2015;Hyttinen et al., 2018;Riva et al., 2019). Therefore, the reported OOM concentration in this study should
be regarded as the lower limit. The concentration of each OOM molecule can be calculated as follows:
$$[OOM] = \frac{\sum_{i=0}^{1}(HNO_3)_i NO_3^-(OOM) + (HNO_3)_i(OOM-H)^-}{\sum_{i=0}^{2}(HNO_3)_i NO_3^-} \times C \div T_{OOM}$$



The numerator on the right-hand side is the sum of detected signal of that OOM in the unit of counts per second
(cps), either as a cluster of a neutral molecule combined with $NO_3^-$, $(HNO_3)_iNO_3^-(OOM)$), or as a deprotonated ion,
$(OOM-H)^-$. It should be pointed out that in nitrate CIMS, most OOMs were detected as the cluster form of $NO_3^-$
(OOM). The denominator is the sum of all reagent ion signals in cps. C is the calibration factor of sulfuric acid,
ranging from $6.07 \times 10^9$ to $7.47 \times 10^9$ cm$^{-3}$ / (normalized cps) during the whole year. Such a narrow range of the
calibration factor also indicated that our instrument had a stable performance during the measurement period. $T_{OOM}$
is the relative transmission efficiency of a specific OOM molecule in comparison with that of the reagent ions.

126       The number concentration of aerosol particles from 6 to 840 nm was measured by a Differential Mobility Particle

Sizer (DMPS) (Aalto et al., 2001). The mass concentration of $PM_{2.5}$ was measured with a Tapered Element
Oscillating Microbalance Dichotomous Ambient Particulate Monitor (TEOM 1405-DF, Thermo Fisher Scientific
Inc, USA). The chemical composition of the $PM_{2.5}$ was obtained from an Aerosol Chemical Speciation Monitor
(ACSM) (Jayne et al., 2000;Drewnick et al., 2005), and PMF analysis was further performed to separate secondary
organic aerosols from primary ones. Meteorological parameters were measured with a weather station (AWS310,
Vaisala Inc.) located on the rooftop of the building. Concentrations of trace gases, including carbon monoxide (CO),
sulfur dioxide ($SO_2$), nitrogen oxides ($NO_x$), and ozone ($O_3$), were monitored using Thermo Environmental
Instruments (models 48i, 43i-TLE, 42i, 49i, respectively).

135       The measurement period covers four seasons of the year 2019, including 135 days in total. Winter, spring,

summer, and autumn periods range from 5$^{th}$ Jan. to 14$^{th}$ Feb., 15$^{th}$ Mar. to 14$^{th}$ Apr., 10$^{th}$ Jul. to 9$^{th}$ Aug., and 19$^{th}$
Oct. to 18$^{th}$ Nov., respectively.
**2.2. A revised workflow for the classification of OOM Sources**

139       A recently developed workflow, based on the molecular composition as well as the up-to-date knowledge of

atmospheric OOM formation chemistry, was used for retrieving their possible sources (Nie et al., 2022) (Xu et al.,
2021). In this approach, mass spectral binning combined with positive matrix factorization (binPMF) (Zhang et al.,
2019) needs to be performed first to extract the factor of monoterpene OOMs. However, as performing binPMF is
time-consuming and not suitable for large data sets as used in this study, we replaced the binPMF step by the criteria
of nC=10, nO$_{eff}\geq$4, and 2$\leq$DBE$\leq$4 (Fig. 1) for the selection of monoterpene OOMs. Such standards were set
based on their reported composition (Ehn et al., 2012;Yan et al., 2016;Jokinen et al., 2014;Boyd et al., 2015;Berndt
et al., 2016;Berndt et al., 2018). Here, nC is the carbon number. nO$_{eff}$ (= nO–2×nN) is the effective oxygen number,
which subtracts the number of oxygen bonded to nitrogen by assuming that all nitrogen atoms are in the form of
nitrate groups ($-ONO_2$) or peroxynitrate nitrate group ($-OONO_2$). To our best knowledge, this is the common case
for all nitrogen-containing compounds formed through the reaction between $RO_2$ and $NO_x$ (Orlando and Tyndall,
2012;Seinfeld and Pandis, 2016). Exceptions are those nitrophenols (Yan et al., 2016;Wang et al., 2019;Song et al.,
2021) that were classified separately (Nie et al., 2022). The reason for choosing nO$_{eff}$ rather than nO is that it better
reflects the oxidation state of closed-shell molecules as well as their parent $RO_2$ radicals. For example, $C_7H_9O_5$





peroxy radical can produce both $C_7H_{10}O_4$ and $C_7H_9O_6N$ when reacting with NO, and all of them have the same $nO_{eff}$
of four. In addition, $nO_{eff}$ considers the influence of nitrogen and represents volatility more directly (Yan et al.,
2020), and thus makes it easier for the volatility comparison among OOMs with different nitrogen atoms.
DBE denotes the double bond equivalence and is calculated as $(2nC+2–nH–nN)/2$, which is the same as the term
degree of unsaturation. The DBE of one OOM molecule is influenced by both its precursor and the oxidation
processes. For example, aromatic VOCs have DBE values no smaller than 4. For their oxidation products, a previous
study has shown that under OH exposures equivalent to approximately 10-15 days in typical atmospheric conditions,
they possess DBE values no smaller than 2 (Garmash et al., 2020). However, reported monoterpene OOMs also
have DBE values the same as those of aromatic OOMs, which makes them difficult to distinguish. According to
laboratory studies, the majority of monomer products from monoterpene oxidation are C10 compounds (Yan et al.,
2020). Measurement results also showed that the concentrations of C10 aromatic VOCs are very low (Zhang et al.,
2017) compared with other C6-C9 ones. Therefore, those C10 OOMs with DBE values of 2 to 4 are likely
monoterpene OOMs. For OOMs with DBE values smaller than 2, neither aromatics nor monoterpenes oxidation
could explain their formation. Hence, the precursors of those OOMs should be the ones without aromatic rings and
have smaller DBE values, such as alkanes, alkenes, and some unsaturated oxygen-containing VOCs (OVOCs).
OOMs with DBE values of 2 are rather complex. Their precursors could be aromatics, aliphatics, or other unknown
sources, and a detailed discussion of the classification criteria could be found in Nie et al. (Nie et al., 2022). By
performing this revised workflow, OOMs were finally divided into five groups: isoprene (IP) OOMs, monoterpene
(MT) OOMs, aromatic OOMs, aliphatic OOMs, and a small amount of undistinguished OOMs (6-9 %) that cannot
be classified into those four types.

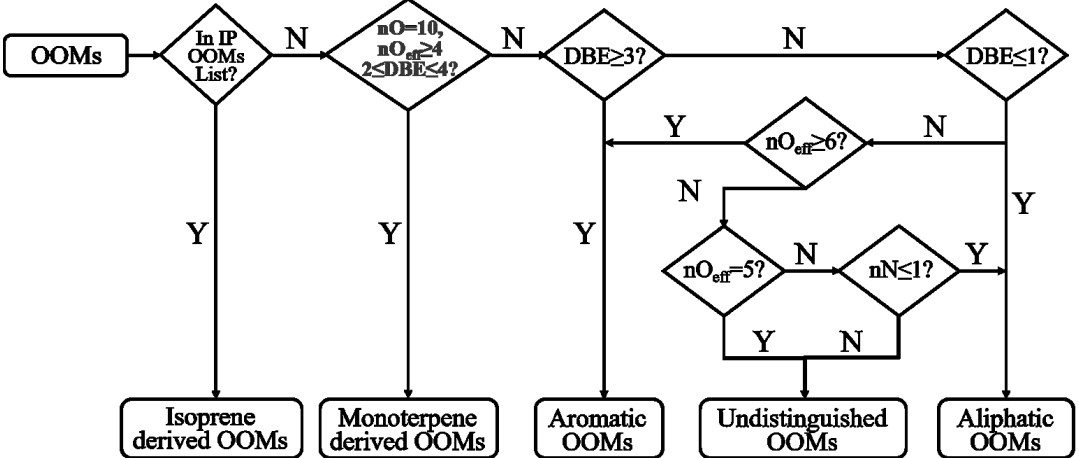

**Figure 1.** Workflow for retrieving OOM sources. "IP OOMs" represents isoprene-derived OOMs. $nO_{eff}$ and nN are the numbers
of effective oxygen and nitrogen in each OOM molecule, respectively. "Y" and "N" denote "Yes" and "No", respectively.



## 3. RESULTS AND DISCUSSIONS

### 3.1. Seasonal Variation of OOM Concentration and Composition

The concentration and molecular composition are the most fundamental characteristics of OOMs. We summarized the OOM concentrations in Beijing and other lower tropospheric sites in Table 1 and Fig. 2 for better comparison. Generally, a clear seasonal trend of OOM concentration in Beijing can be observed, where total OOM concentration is highest in summer ($1.6 \times 10^8$ cm$^{-3}$), followed by autumn ($7.9 \times 10^7$ cm$^{-3}$) and spring ($5.7 \times 10^7$ cm$^{-3}$), and the lowest in winter ($2.3 \times 10^7$ cm$^{-3}$). This apparent increase of OOM concentrations with an increased temperature and theoretical global radiation indicates that elevated solar radiation along with higher temperature favors the generation of OOMs. In comparison to other locations, the level of OOMs in urban Beijing varied within the ranges of previously reported ones (Yan et al., 2016;Roldin et al., 2019;Bianchi et al., 2017;Zha et al., 2018;Huang et al., 2020;Mutzel et al., 2015;Massoli et al., 2018;Nie et al., 2022). Interestingly, the above clear correlation between global radiation (or temperature) and OOM concentration can also be seen in other locations, yet OOMs in forest environments are in general higher than in urban or suburban areas. On the one hand, this observation suggests that OOM formation at a specific environment is prevailingly influenced by the strength of atmospheric photochemistry; on the other hand, forest environment appears to have more abundant OOMs than urban environments, possibly because the OOM yield of biogenic VOCs is higher than that of anthropogenic VOCs (Berndt et al., 2016;Teng et al., 2017;Garmash et al., 2020;Molteni et al., 2018). Yet, a quantitative explanation of the OOM variation between seasons and locations requires comprehensive measurements as well as analyses on both the production and loss of OOMs.

**Table 1.** Mean, standard deviation (Std), median, 25 and 75 percentiles (25[th] and 75[th]) of measured OOM concentrations at various lower tropospheric sites.

| Measurement Site | Period | Mean (cm$^{-3}$) | Std (cm$^{-3}$) | Median (cm$^{-3}$) | 25[th] (cm$^{-3}$) | 75[th] (cm$^{-3}$) | Reference |
|---|---|---|---|---|---|---|---|
| Beijing, China | 2019 Jan.-Feb. | $2.7 \times 10^7$ | $1.7 \times 10^7$ | $2.3 \times 10^7$ | $1.3 \times 10^7$ | $3.6 \times 10^7$ | This study |
| Beijing, China | 2019 Mar.-Apr. | $6.9 \times 10^7$ | $5.1 \times 10^7$ | $5.7 \times 10^7$ | $3.1 \times 10^7$ | $8.9 \times 10^7$ | This study |
| Beijing, China | 2019 Jul.-Aug. | $1.6 \times 10^8$ | $7.5 \times 10^7$ | $1.6 \times 10^8$ | $1.1 \times 10^8$ | $2.2 \times 10^8$ | This study |
| Beijing, China | 2019 Oct.-Nov. | $8.3 \times 10^7$ | $5.2 \times 10^7$ | $7.9 \times 10^7$ | $4.0 \times 10^7$ | $1.2 \times 10^8$ | This study |
| Hong Kong, China | 2018 Nov. | $2.3 \times 10^8$ | $1.1 \times 10^8$ | $2.1 \times 10^8$ | $1.5 \times 10^8$ | $2.9 \times 10^8$ | 2022, Nie et al. |
| Shanghai, China | 2018 Nov. | $7.8 \times 10^7$ | $6.3 \times 10^7$ | $6.1 \times 10^7$ | $2.6 \times 10^7$ | $1.2 \times 10^8$ | 2022, Nie et al. |
| Nanjing, China | 2018 Nov. | $7.7 \times 10^7$ | $5.4 \times 10^7$ | $7.2 \times 10^7$ | $3.1 \times 10^7$ | $1.1 \times 10^8$ | 2022, Nie et al. |
| Hyytiälä forest, Finland | 2012 Apr.-May | $7.5 \times 10^7$ | $6.1 \times 10^7$ | $5.7 \times 10^7$ | $3.6 \times 10^7$ | $9.2 \times 10^7$ | 2016, Yan et al. |
| Hyytiälä forest, Finland | 2013 May | $1.4 \times 10^7$ | $7.9 \times 10^6$ | $1.2 \times 10^7$ | $8.1 \times 10^6$ | $1.8 \times 10^7$ | 2019, Roldin et al. |
| Hyytiälä forest, Finland | 2013 Apr.-Jun. | $4.5 \times 10^7$ | $1.2 \times 10^7$ | $4.9 \times 10^7$ | $3.2 \times 10^7$ | $5.5 \times 10^7$ | 2017, Bianchi et al. |
| Hyytiälä forest, Finland | 2016 Sep. | $1.2 \times 10^8$ | $1.0 \times 10^8$ | $8.0 \times 10^7$ | $3.8 \times 10^7$ | $1.7 \times 10^8$ | 2018, Zha et al. |
| Melpitz, Germany | 2013 Jul. | $2.7 \times 10^8$ | $1.7 \times 10^8$ | $2.5 \times 10^8$ | $1.4 \times 10^8$ | $3.5 \times 10^8$ | 2015, Mutzel et al. |
| Alabama forest, USA | 2013 Jun.-Jul. | $4.7 \times 10^9$ | $1.5 \times 10^9$ | $4.8 \times 10^9$ | $3.7 \times 10^9$ | $5.3 \times 10^9$ | 2018, Massoli et al. |

To further demonstrate the seasonal influence of solar radiation and precursor VOCs on OOMs concentration, we classified OOMs of each season into four groups based on the brightness parameter (Dada et al., 2017) and

$PM_{2.5}$ level, respectively. As shown in Fig. S2, in seasons other than summer, OOM concentration under polluted
conditions is much higher than that under clean conditions, which likely results from the elevation of precursors
coming along with polluted air masses and the accumulation during the pollution. Besides, the concentration of
total OOMs on sunny days is higher than that on cloudy days, implying that photochemical oxidation plays a key
role in the production of OOM molecules. There is one exception that OOM concentration is not significantly
different between sunny and cloudy days under clean condition in autumn, and the cause cannot be concluded in
this study without a complete VOC measurement.

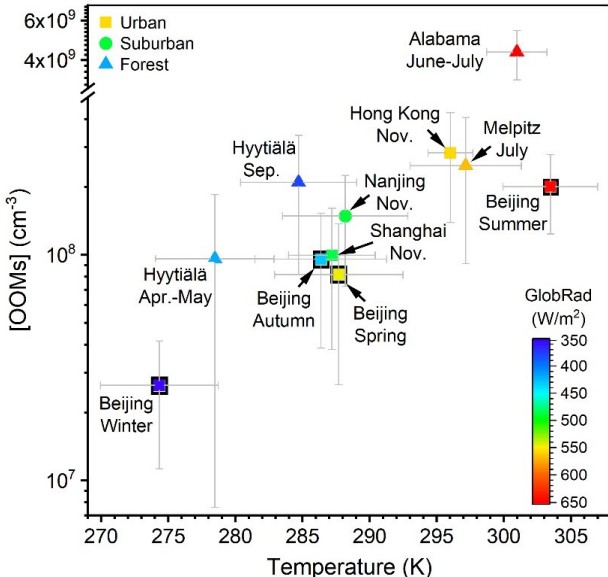

**Figure 2.** OOM concentration vs. temperature at various lower tropospheric sites during daytime (07:00 – 17:00). Data points
are colored by theoretical global radiation (GlobRad). Square, circle, and triangle markers represent urban, suburban, and forest
areas, respectively. The gray error bars show standard deviations (1σ). Nanjing, Shanghai and Hong Kong datasets are from
Nie et al. (Nie et al., 2022), Melpitz data is from Mutzel (Mutzel et al., 2015), Alabama data is from Massoli et al. (Massoli et
al., 2018), and Hyytiälä datasets are from Yan et al. (Yan et al., 2016) and Zha et al (Zha et al., 2018).
For OOM composition, the two-dimensional H/C-$O_{eff}$/C (ratio of hydrogen number to carbon number vs. ratio
of effective oxygen number to carbon number) diagrams are plotted to show its characteristics (Fig. S3). And main
CHO, CHON, and $CHON_2$ OOM species are also summarized in Table 2. Generally, the composition of OOM
molecules exhibits high similarity among different seasons, suggesting no significant changes in OOM formation
in general. However, two seasonal characteristics can be found. First, the most oxygenated OOM molecules, such
as $C_nH_{2n-2}O_{6,7}$, $C_nH_{2n-4}O_{7,8}$ $C_nH_{2n+1}O_8N$ and $C_nH_{2n-1}O_9N$ OOMs, are mainly observed in summer, and meanwhile,
the least oxygenated ones, e.g., $C_nH_{2n-7}O_2N$ and $C_nH_{2n-9}O_{4,5}N$ are mostly detected in winter. These observations
indicate that, in addition to the enhanced OOM concentration, strong photochemistry also leads to a high oxidation
state of OOM. And these summer-specific OOMs can be classified as highly oxygenated organic molecules (HOMs).
Second, $C_5H_{10}O_8N_2$ is exceedingly high in summer. A previous study suggested that $C_5H_{10}O_8N_2$ is one dominant





oxidation product from isoprene (Xu et al., 2021), and therefore, the high concentration of $C_5H_{10}O_8N_2$ is a clear
indication of intensive isoprene oxidation in summer, which will be discussed in detail in Sect. 3.2.

**Table 2.** Main CHO, CHON and CHON$_2$ OOM species measured in this study.

| DBE | CHO OOMs | CHON OOMs | CHON$_2$ OOMs |
|---|---|---|---|
| 0 | $C_nH_{2n+2}O_6$ | $C_nH_{2n+1}O_{3-8}N$ | $C_nH_{2n}O_{4-11}N_2$ |
| 1 | $C_nH_{2n}O_{2-8}$ | $C_nH_{2n-1}O_{3-9}N$ | $C_nH_{2n-2}O_{4-10}N_2$ |
| 2 | $C_nH_{2n-2}O_{3-7}$ | $C_nH_{2n-3}O_{3-9}N$ | $C_nH_{2n-4}O_{5-11}N_2$ |
| 3 | $C_nH_{2n-4}O_{2-8}$ | $C_nH_{2n-5}O_{3-10}N$ | $C_nH_{2n-6}O_{8-11}N_2$ |
| 4 | $C_nH_{2n-6}O_{3-9}$ | $C_nH_{2n-7}O_{3-9}N$ | $C_nH_{2n-8}O_{7,8}N_2$ |
| 5 | $C_nH_{2n-8}O_{3-8}$ | $C_nH_{2n-9}O_{4-10}N$ | $C_nH_{2n-10}O_{6-10}N_2$ |


For a better understanding of OOM composition variation among seasons, the distributions of nC, nO$_{eff}$, nN, and
DBE, as well as their seasonal variations, are further analyzed. It should be pointed out that the concentration
($5.3\times10^7$ cm$^{-3}$) and the fraction (33 %) of IP OOMs in summer is much higher than those in other three seasons, and
therefore they are plotted in bars with diagonal lines individually (Fig. 3). In terms of carbon content, the majority
of OOMs contain 5 to 10 carbon atoms. For OOMs with 6 to 10 carbon atoms, in seasons other than summer, C6
are the most abundant, and a decreasing trend can be seen along with an increasing nC, while in summer, an opposite
trend is observed, i.e., the relative contribution increases with an increasing nC. The causes behind the different
trends in summer and other seasons are complex but might include changes in the precursor VOC distribution,
varying reactivity responses of VOCs to temperature, and the volatilities of OOMs that influence their atmospheric
lifetime. Further analysis on this topic will be made in the future. For C5 OOMs, the contribution from isoprene
varies from less than half in winter and spring to ~70 % in summer. The high contribution of IP OOMs in summer
is in line with the strong isoprene emission coupled with the enhanced photo-oxidation (Cheng et al., 2018;Zhang
et al., 2020).
Concerning the oxygen content, most OOMs contain 3 to 7 effective oxygen atoms, accounting for 86-95 % of
total OOMs in all seasons. With the increase of effective oxygen number, the contribution of corresponding OOMs
first increases and then decreases, with nO$_{eff}$=4 OOMs having the highest fraction. Fig. S4 shows that the
concentration-weighted average nO$_{eff}$ is the highest in summer and lowest in winter and autumn, which is consistent
with the observation of individual molecules where the most highly oxygenated ones are usually more abundant in
summer. The enhanced multi-step oxidation (Garmash et al., 2020;Wu et al., 2021) and favored auto-oxidation
(Molteni et al., 2018;Wang et al., 2017;Wang et al., 2018b;Bianchi et al., 2019) at high temperatures in warmer
seasons are most likely the causes. Furthermore, when taking IP OOMs into account, nO$_{eff}$=4 becomes even more
prominent in summer, in which $C_5H_{10}O_8N_2$ takes the largest portion and accounts for 77 % of nO$_{eff}$=4 IP OOMs. In
winter and autumn, however, nO$_{eff}$=3 has a much higher fraction than those in the other two seasons, and those
OOMs are mainly composed of low-DBE compounds, such as $C_nH_{2n}O_7N_2$, $C_nH_{2n-2}O_7N_2$, and $C_nH_{2n-1}O_5N$ species.
As for nitrogen content, the vast majority (98-99 %) of OOMs contain 0 to 2 nitrogen atoms, in which CHON
OOMs take the largest fraction, varying from 42 % to 51 % among seasons. It should be noted that, although the



mixing ratios of $NO_x$ in different seasons change significantly, the nitrogen distributions of non-isoprene OOMs are
similar, which is probably due to the fact that NO (0.6-10.0 ppbv) and $NO_2$ (8.9-28.3 ppbv) concentrations in urban
Beijing are always high throughout the year. Those nitrogen atoms could come from either $NO_3$ radical oxidation
or $NO_x$ termination. During the day, nitrogen is likely added mainly through $NO_x$ termination as $NO_3$ radical should
be photolyzed or titrated by NO.  However, in the absence of $NO_3$ photolysis at night, when NO concentration is
low, the concentration of $NO_3$ radical could reach up to ~10 pptv ($2.7 \times 10^8$ cm$^{-3}$) in Beijing (Wang et al., 2018a).
Under such levels, the $NO_3$ radial could even dominate the oxidation of biogenic VOCs (i.e., isoprene and
monoterpenes) and some aliphatic VOCs, yet oxidation of aromatic VOCs are driven by OH radicals (Table S4
and Table S5). Therefore, nitrogen is possibly added through both processes at night.

265       In the case of DBE distribution, most OOMs comprise 0 to 6 DBE values and there is not too much difference

among seasons for non-isoprene OOMs. Generally, with the increase of DBE, the fraction of corresponding non-
isoprene OOMs first increases and then decreases, with DBE=2 OOMs having the highest contribution (20-29 %).
And this is possibly caused by the fact that almost all precursor VOCs, such as aromatics, aliphatics and
monoterpenes, can form oxidation products with DBE value of 2. OOMs with DBE larger than 4 and nC no smaller
than 10 are likely derived from polycyclic aromatic hydrocarbons (PAHs, DBE≥7), and their fraction varies from
5 % in summer to 7-8% in the other three seasons. This demonstrates that PAHs may also have a non-negligible
contribution to total OOMs. For IP OOMs, most of them possess 0 or 1 DBE, and only a small fraction (2-16 %) of
them retains DBE of precursor isoprene. This means that isoprene oxidation mostly causes a DBE reduction, such
as via OH addition to the double carbon bond and the formation of hydroxyl or hydroperoxyl groups; on the contrary,
DBE-augment processes, such as the formation of carbonyl and epoxide, are not facilitated. Specifically, in summer,
when $C_5H_{10}O_8N_2$ is exceptionally high, the fraction of DBE=0 IP OOMs reaches 75 %, suggesting that few double
bond is retained and that the formation of carbonyl or epoxide groups is of less importance in the isoprene oxidation.

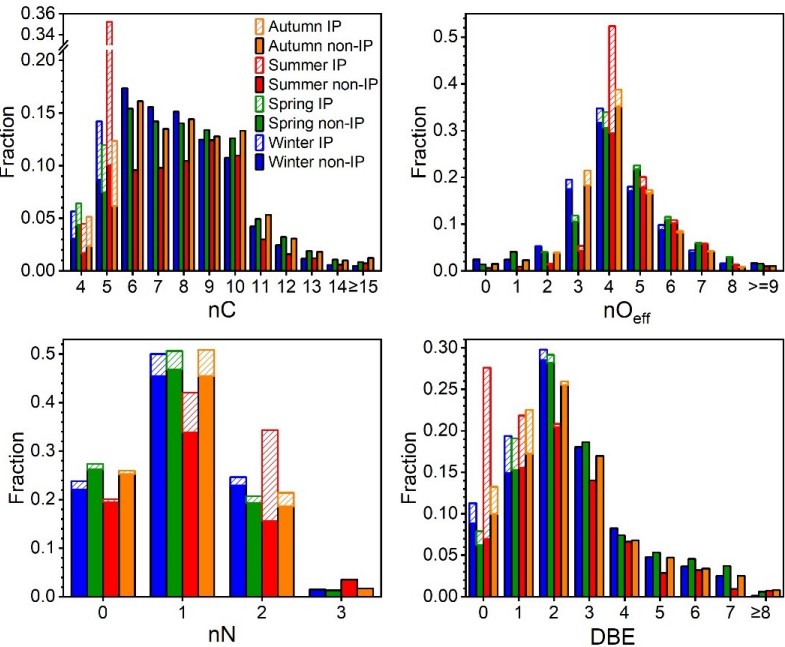

**Figure 3.** Number of carbon (nC), effective oxygen ($nO_{eff}$), nitrogen (nN), and double bond equivalence (DBE) distribution of OOMs for four seasons. The abbreviations "IP" and "non-IP" represent IP OOMs and other non-isoprene OOMs, respectively. The bars with diagonal lines and filled colors represent IP OOMs and non-isoprene OOMs, respectively.

### 3.2. Characteristics of Source-classified OOMs

With the workflow described in Sec.2.2, total OOMs were classified into four types: IP OOMs, MT OOMs, aromatic OOMs, and aliphatic OOMs. As shown in Fig. 4, the seasonal concentrations of OOMs from different sources vary with the same trend, highest in summer, followed by autumn, spring, and winter. During the whole year, aromatic OOMs (29-41 %) and aliphatic OOMs (26-41 %) are the most abundant categories, demonstrating that OOMs in Beijing are dominantly from anthropogenic sources. This is also consistent with the observation of SOA composition in previous studies (Le Breton et al., 2018;Mehra et al., 2021). In terms of OOMs from biogenic sources, IP OOMs show a prominent contribution in summer (33 %), which is much higher than those in other seasons (8-10 %). Although it is recently suggested that isoprene can have both biogenic and anthropogenic sources (Wagner and Kuttler, 2014;Panopoulou et al., 2020), the much higher enhancement of IP OOMs in summer can only be explained by the large additional biogenic emission (Cheng et al., 2018;Mo et al., 2018). For MT OOMs, however, the fractional contribution does not show a seasonal variation as clear as that of IP OOMs – it only varies between 5 % and 6 %.



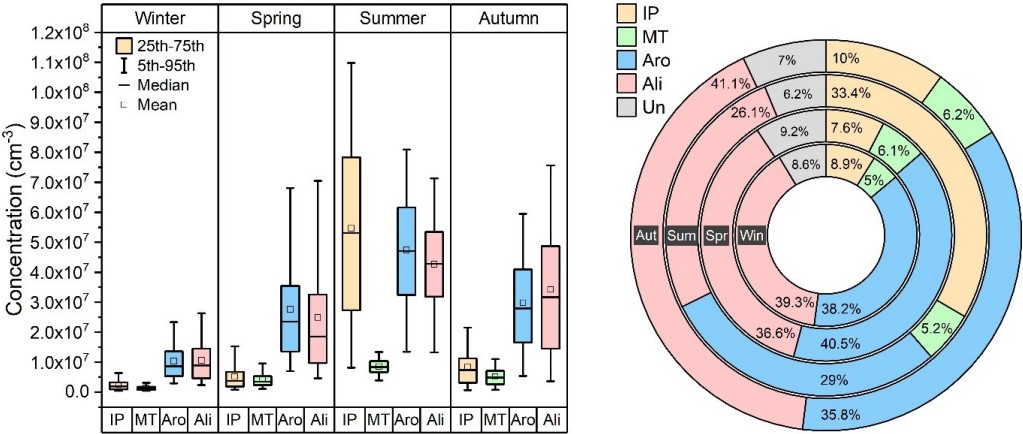

**Figure 4.** Concentration (left panel) and fraction (right panel) of source-classified OOMs in four seasons. The abbreviations "IP" "MT" "Aro" "Ali" and "Un" stand for IP OOMs, MT OOMs, aromatic OOMs, aliphatic OOMs and undistinguished OOMs respectively. "Win" "Spr" "Sum" and "Aut" represent winter, spring, summer and autumn separately.

### 3.2.1 Characteristics of biogenic OOMs

The spectral profiles and the fractions of IP OOMs with different nitrogen numbers in four seasons are shown in Fig. 5. Prominent IP OOM species include $C_4H_6O_4$, $C_5H_8O_4$, $C_4H_7O_{6,7}N$, $C_5H_9O_{5-7}N$, $C_5H_{10}O_{7,8}N_2$ and $C_5H_9O_{10}N_3$. $CHON_3$ OOM ($C_5H_9O_{10}N_3$) is detected in all four seasons, suggesting that multi-generation oxidation is involved throughout the year. Besides, the composition of IP OOMs exhibits clear seasonal variation. First, compared with other three seasons, $C_5H_{10}O_8N_2$ and $C_5H_9O_{10}N_3$ have much higher contributions in summer, indicating that $NO_x$ may be involved more efficiently in the oxidation process of isoprene despite its lower concentration (Fig. S6 and Table S1). In addition, nighttime $NO_3$ radicals produced efficiently during summer nights (Wang et al., 2018a) should also promote their formation. Second, despite the overall lowest concentrations of IP OOMs in winter, $C_4H_8O_7$ exhibits a maximum concentration (Fig. S5) and the highest fraction, implying that it may has additional sources other than isoprene oxidation in winter. Third, due to the influence of $C_5H_{10}O_8N_2$ and $C_5H_9O_{10}N_3$, $CHON_2$ and $CHON_3$ IP OOMs take extremely large proportion (~ 67 %) in summer. And interestingly, the seasonal trend of nitrate IP OOM fraction (from largest to smallest is summer, autumn, spring and winter) did not follow the variation of $NO_x$ concentration (from highest to lowest is autumn, winter, spring and summer, Fig. S1 and Table S1), which suggests that the formation of nitrate IP OOMs probably has a non-linear response to $NO_x$.

The concentrations of these prominent IP OOM molecules during summertime in Beijing, Nanjing (32.12° N, 118.95° E) (Liu et al., 2021) and Alabama mixed-forest (32.90° N, 87.25° W) (Krechmer et al., 2015;Massoli et al., 2018) are further compared (Fig. 6). Please note that these molecules plotted are not all the IP OOMs but rather selected abundant ones reported in the literature and in this study. As shown in Fig. 6 (A), IP OOMs exhibit the highest concentration in Alabama forest and the lowest one in Beijing, and the level of IP OOMs in Beijing and Nanjing are comparable. This concentration difference is likely caused by the variation of biogenic isoprene





emissions, since Alabama measurement was conducted in a forest, Nanjing site is a suburban area with large
vegetation coverage nearby, and Beijing site is located in urban downtown. Besides, the overall varying patterns of
IP OOM species in Beijing and Nanjing are very similar, indicating that isoprene in those two urban sites undergo
similar oxidation pathways. In terms of oxygen distribution, Beijing and Nanjing are rather similar in that $nO_{eff}=4$
OOMs contribute the most, whereas in Alabama $nO_{eff}=5$ ones are the most abundant (Fig. 6 (B)). This lower oxygen
content in urban cities is probably caused by the high $NO_x$ levels (11.1 ppbv, 8.5 ppbv and 0.5 ppbv for Beijing,
Nanjing and Alabama respectively, Fig. S6), since $NO_x$ efficiently suppresses the oxygen addition of $RO_2$ radicals
(Zhao et al., 2018). Furthermore, the different $NO_x$ levels among the three sites also influence the nitrogen content
that Beijing is the highest and Alabama is the lowest (Fig. 6 (C)).
From the perspective of diurnal variation, CHO IP OOMs in Beijing possess one daytime peak, while CHON
OOMs mainly contain day-night-dual-peak or nocturnal-peak-only types (Fig. S7), which is similar to that reported
in Alabama forest (Massoli et al., 2018). But it should be noted that the diurnal variations of some CHON IP OOMs
with same molecular composition in this study and Massoli et al. (2018) are not identical, suggesting that their
formation pathways are different under various atmospheric conditions.

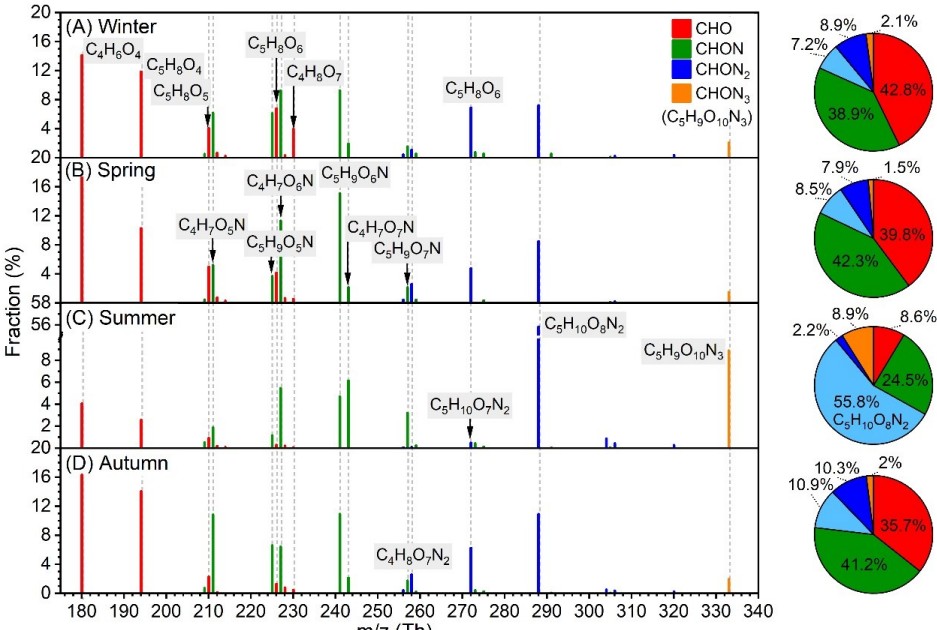


**Figure 5.** Fractional profiles of each IP OOM molecule in (A) winter, (B) spring, (C) summer and (D) autumn. The mass to
charge ratio (m/z) denotes OOM clustered with $NO_3^-$ or as deprotonated ones. The red, green, blue and orange bars are for
CHO, CHON, $CHON_2$ and $CHON_3$ OOMs respectively. Please note that $CHON_3$ OOMs only include $C_5H_9O_{10}N_3$.



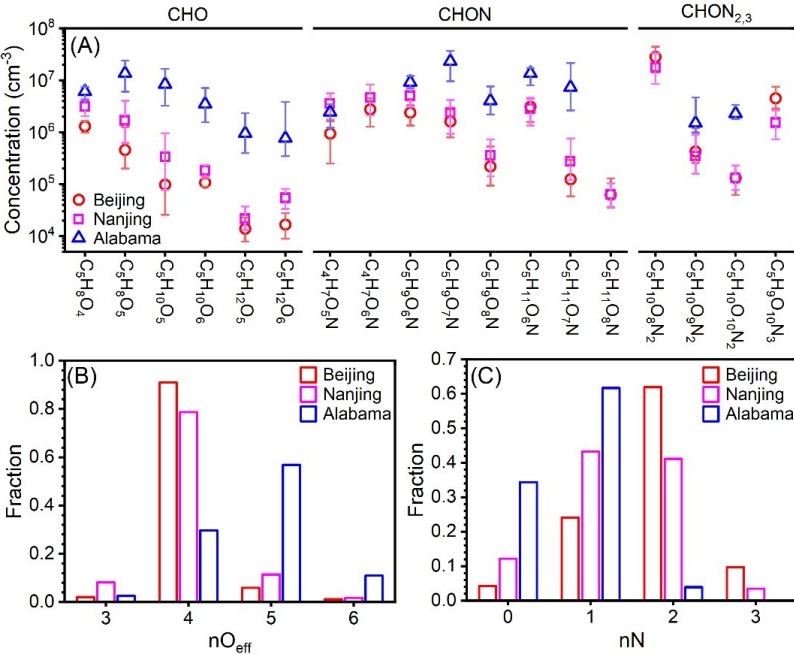

**Figure 6.** (A) Concentration comparison of specific fingerprint IP OOM molecules between our study and previously reported ones (Krechmer et al., 2015;Massoli et al., 2018;Liu et al., 2021). The markers are median concentration values, and the upper and lower range of the error bar denote 25th and 75th percentiles respectively. Distribution of (B) effective oxygen number, $nO_{eff}$, and (C) nitrogen number, nN, of IP OOMs plotted in figure (A).

Different from IP OOMs, the overall composition distributions of MT OOMs in the four seasons are quite similar and vary with identical oxygen addition patterns (Fig. 7). Predominant MT OOM molecules are $C_{10}H_{14}O_{4-8}$, $C_{10}H_{16}O_{4-6}$, $C_{10}H_{18}O_{4,5}$, $C_{10}H_{13,15,17}O_{6-9}N$ and $C_{10}H_{14,16}O_{8-10}N_2$. Besides, most MT OOMs belong to CHON category (54-59 %), and the CHO (18-21 %) and $CHON_2$ (19-27 %) ones, with a comparable contribution during the year.

Then, for a better understanding of MT OOM characteristics under different atmospheric environments, representative MT OOM molecules in summer Beijing, spring Hyytiälä forest (61.8° N) (Yan et al., 2016) and summer Alabama mixed-forest (Massoli et al., 2018) are further compared. The following differences can be identified. First, MT OOM concentrations are the highest in Alabama and the lowest in urban Beijing (Fig. 8 (A)), which should result from the synergetic influence of UVB, temperature and precursor monoterpenes. Second, the levels of the two MT radicals that have high concentrations in the Hyytiälä forest, $C_{10}H_{15}O_8\cdot$ and $C_{10}H_{15}O_{10}\cdot$, are not detected in Beijing. This is possibly caused by both low monoterpene abundance (Cheng et al., 2018) and high $NO_x$ concentration in Beijing (11.06 ppbv, Fig. S6), which lead to a low production rate and high loss rate of $RO_2$ radicals. Third, most MT OOMs in urban Beijing possess 5 or 6 effective oxygen, whereas in the forest environment a large fraction of them can hold 7 to 10 effective oxygen (Fig. 8 (B)). This again suggests that high $NO_x$ in Beijing effectively inhibits the oxygen addition processes (Zhao et al., 2018;Orlando and Tyndall, 2012). Additionally, high $NO_x$ in Beijing also leads to high nitrogen content (Fig. 8 (C)) by promoting the termination reaction between $RO_2$





and NO$_x$ (Orlando and Tyndall, 2012) and facilitating the formation of NO$_3$ radical (Wang et al., 2018a), which
further leads to the formation of nitrate MT OOMs (Boyd et al., 2015;Nah et al., 2015).

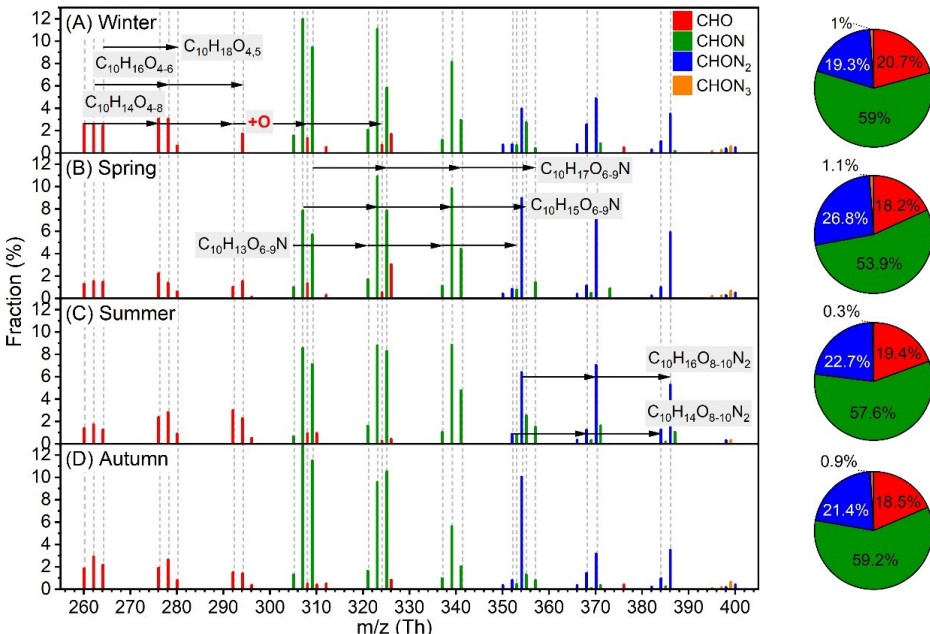

**Figure 7.** Fractional profiles of each MT OOM molecule in (A) winter, (B) spring, (C) summer and (D) autumn. The mass to
charge ratio (m/z) denotes OOM clustered with NO$_3^-$ or as deprotonated ones. The red, green, blue and orange bars are for
CHO, CHON, CHON$_2$ and CHON$_3$ OOMs respectively.

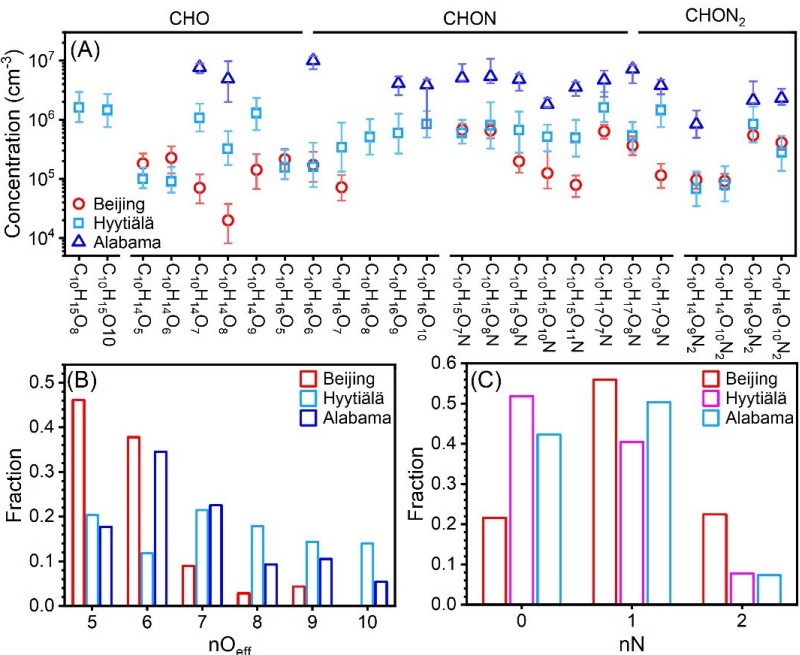

**Figure 8.** (A) Concentration comparison of specific fingerprint MT OOM molecules between our study and previously reported
ones (Massoli et al., 2018;Yan et al., 2016). The markers are median concentration values, and the upper and lower range of
the error bar denote $25^{th}$ and $75^{th}$ percentiles respectively. Please note that only the summer Beijing data was plotted as the
overall pattern of MT OOMs in Beijing summer and spring are very similar (Fig. 7). Distribution of (B) effective oxygen
number, $nO_{eff}$, and (C) nitrogen number, $nN$, of MT OOMs plotted in figure (A).

**3.2.2 Characteristics of anthropogenic OOMs**
Although the oxidation pathways and product composition of a few aromatic VOCs have been studied previously,
the reported products are of much less diversity compared to the complex real atmosphere. Therefore, we rely on
the workflow (see Sect. 2.2 and Nie et al. 2022) to find out possible aromatic OOMs in our measurement. Among
the deduced aromatic OOMs, almost all C6-C9 CHO and C6 CHON compounds have been detected in previous
benzene, toluene, xylene, ethylbenzene and mesitylene experiments (Molteni et al., 2018;Garmash et al., 2020) (see
detail in Table S8), which demonstrates the reliability of our workflow.
As shown in Fig. 9 (A) to (D), predominant aromatic species in different seasons possess high similarity and
they could be classified into $C_{6-9}H_{8-14}O_4$, $C_{6-8}H_{8-12}O_5$, $C_{7,8}H_{8,10}O_5$, $C_{6-10}H_{7-15}O_6N$, $C_{7-9}H_{9-13}O_7N$, $C_{8-10}H_{11-15}O_8N$ and
$C_{8,9}H_{12-14}O_{10}N_2$ categories, among which a prominent $CH_2$ spacing is seen. Such patterns are most likely due to the
co-existence of homologous precursor VOCs, although fragmentation processes during the oxidation could also
play a role (Pan and Wang, 2014;Zaytsev et al., 2019;Xu et al., 2020). Besides, the distribution of CHON aromatic
OOMs in winter and autumn are very similar, with $C_8H_{11}O_6N$ being the highest; in comparison, the overall
distribution moves to higher oxygen content in summer, e.g., $C_8H_{11}O_7N$ becomes the largest one. This suggests that
enhancement of radiation, which leads to strong photochemistry and high temperature, and the reduction of $NO_x$ in





summer benefit the formation of highly oxygenated organic molecules (Garmash et al., 2020;Orlando and Tyndall,
2012). There are also fingerprint molecules for different seasons, such as $C_{10}H_8O_6$ in spring, $C_9H_{10}O_{11}N_2$ in summer,
and $C_{10}H_6O_4$ in summer and autumn. Due to the complexity of real atmosphere, the reason for their seasonal
variation is unclear, and further analysis is warranted. In terms of nitrogen content, aromatic OOMs contain large
fraction of CHO (42-52 %) and CHON (39-51 %) species. The contribution of $CHON_2$ OOMs reaches the highest
in summer, which again indicates that the involvement of $NO_x$ is enhanced under the influence of elevated UVB
and temperature. The carbon distribution among seasons are very similar (Fig. S9), where C4 to C9 aromatic OOMs,
probably derived from monocyclic aromatic hydrocarbons, make up 68-76 %, and other C≥10 ones, of which 59-
68 % are likely the oxidation products from PAHs (DBE≥5) (Table S9), take up 24-32 %. This implies that the
relative abundance of emitted aromatic precursors with different carbon atoms is quite stable during the year.

400       Major aliphatic OOM molecules in different seasons are highly similar, and they possess more evident

homologous patterns than aromatic OOMs (Fig. 9 (E) to (H) and Table S10). The dominant species of aliphatic
OOMs are $C_{6-9}H_{10-16}O_4$, $C_{6-10}H_{11-19}O_6N$, $C_{5-10}H_{7-17}O_6N$, $C_{5-10}H_{8-18}O_8N_2$ and $C_{6-9}H_{12-18}O_7N_2$, and some less oxygenated
compounds, e.g., $C_{6-8}H_{11-15}O_5N$, also have considerable contributions in winter and autumn. For nitrogen content,
CHO aliphatic OOMs take smaller fractions in all seasons (8-11 %) in comparison to aromatic CHO OOMs (42-
52 %). This implies that $NO_x$ termination may dominate the formation of aliphatic close-shell molecules, or that the
branching ratio of the aliphatic $RO_2$ - $NO_x$ reaction is higher than that of the aromatic one. Besides, unlike aromatic
OOMs, aliphatic $CHON_2$ OOMs have a bigger contribution in winter than in the other three seasons. This is because
a major sequence of $CHON_2$ OOMs, $C_{6-14}H_{12-28}O_7N_2$, is found to be coincided with $PM_{2.5}$ (Fig. S10), which is
frequently high in winter. $C_{6-14}H_{12-28}O_7N_2$ OOMs are classified as either SVOCs or IVOCs (Table S11). Therefore,
such a good correlation indicates that $C_{6-14}H_{12-28}O_7N_2$ OOMs themselves, or their precursor VOCs, are able to
transport long distance along with $PM_{2.5}$. It is very likely that they are equilibrated in larger gas-phase concentrations
as SOA also increases with the elevation of $PM_{2.5}$ (Fig. S11). Those pollution-related OOMs take the largest (14 %)
and smallest (2 %) fraction in winter and summer respectively (Table S9). In terms of carbon distribution, there is
not too much difference among seasons, in which the relatively short C4 to C9 aliphatic OOMs make up 83-90 %
and the longer ones take up 10-17 % (Fig. S9).



**Figure 9.** Fractional profiles of each aromatic OOM molecule in (A) winter, (B) spring, (C) summer and (D) autumn, and of each aliphatic OOM molecule in (E) winter, (F) spring, (G) summer and (H) autumn. The mass to charge ratio (m/z) denotes OOM clustered with $NO_3^-$ or as deprotonated ones. The red, green, blue and orange bars are for CHO, CHON, $CHON_2$ and $CHON_3$ OOMs respectively. Species marked with grey dashed lines are primary ones during the year, and species in grey background are special ones for specific seasons.



Up till now, field measurements of anthropogenic OOMs are rare (Liu et al., 2021;Nie et al., 2022). In general,
the concentrations of aromatic and aliphatic OOMs in Beijing are comparable with those in other Chinese
megacities (Table S12), and fingerprint aromatic and aliphatic OOM molecules in Beijing and Nanjing are also
identical (Table S13). This suggests that the OOM production, including both the precursor emissions and oxidation
mechanisms, may share high similarities in megacities. Yet, a more systematic comparison can only be made when
measurements at more locations are available in the future.

### 3.3. Atmospheric implication: OOM contribution to SOA through condensation

The volatility of organic compound determines its partitioning between gas and particle phases, and thus
influences its atmospheric lifetime, gas-phase concentration, and contribution to SOA. Therefore, we analyze the
characteristics of OOM volatility and summarize the results in Fig. 10. The seasonal variations of OOMs classified
as ELVOCs (extremely low-volatility organic compounds), LVOCs (low-volatility organic compounds) and
SVOCs (semi-volatile organic compounds) follow the same trend as that of total OOMs, with the highest
concentrations in summer ($1.3\times10^7$ cm$^{-3}$, $4.0\times10^7$ cm$^{-3}$ and $8.4\times10^7$ cm$^{-3}$ for ELVOCs, LVOCs, and SVOCs,
respectively) and the lowest ones in winter ($4.4\times10^6$ cm$^{-3}$, $9.4\times10^6$ cm$^{-3}$ and $5.3\times10^6$ cm$^{-3}$ for ELVOCs, LVOCs, and
SVOCs, respectively). Here, we focus particularly on OOMs with relatively low volatility with high potential
contributing to the formation of SOA.
Due to the concentration variation of four source-classified OOMs and their temperature-dependent volatility
distribution (Table S15), their fractions within different volatility ranges have distinct seasonal characteristics (Fig.
10 (B)). Among ELVOCs, aromatic OOMs take the largest fractions, ranging from 72 to94 % throughout the year.
For LVOCs, aromatic (34-51 %) and aliphatic OOMs (17-42 %) are the two that have the largest proportions. And
MT OOMs, favored by its low volatility (Table S15) (Tröstl et al., 2016;Yan et al., 2020), also take up 14 % of
LVOCs in seasons other than winter. IP OOMs, however, due to its high volatility (Table S15) (Krechmer et al.,
2015;Xu et al., 2021), do not have an appreciable contribution to ELVOCs and LVOCs even in summer when its
concentration is exceedingly high. Consequently, it is likely that the pure condensation of IP OOMs has little
contribution to SOA growth regardless of the season.

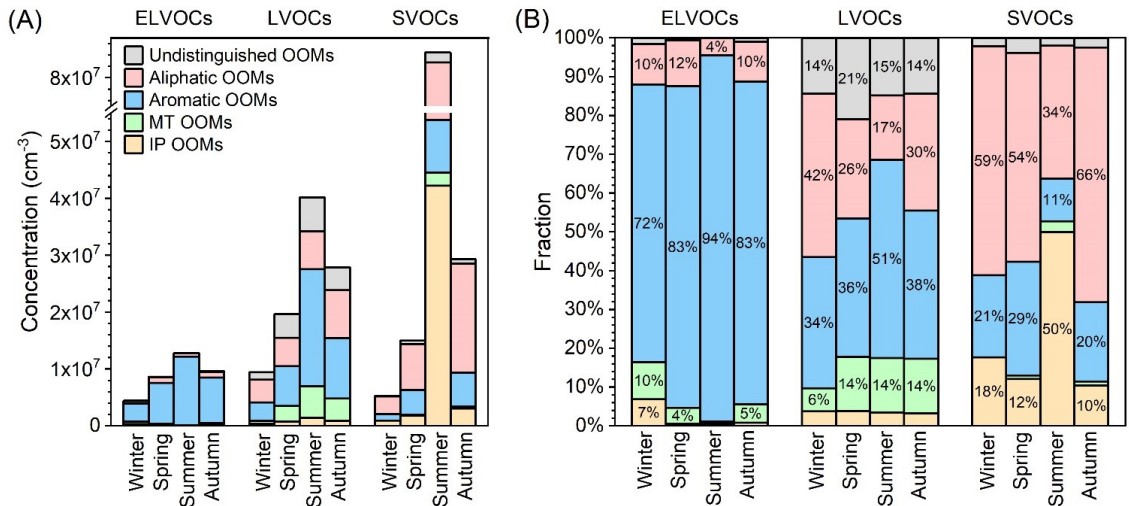

**Figure 10.** (A) Concentration and (B) fraction of source-classified OOMs in ELVOCs (extremely low volatile organic compounds), LVOCs (low volatile organic compounds) and SVOCs (semi-volatile organic compounds) in four seasons. Please also note that the fractions smaller than 4% are not marked.

The rate of OOM condensation onto particles, referred to as condensation flux hereafter, was calculated based on the particle dynamic model proposed by Trostl et al. (Tröstl et al., 2016) (see details in Sec. S2). In terms of seasonal variation (Fig. 11 (A)), OOM condensation flux exhibits the highest level in autumn (0.64 $\mu g \cdot m^{-3} \cdot h^{-1}$), followed by summer (0.61 $\mu g \cdot m^{-3} \cdot h^{-1}$), spring (0.41 $\mu g \cdot m^{-3} \cdot h^{-1}$), and decreases to the lowest in winter (0.30 $\mu g \cdot m^{-3} \cdot h^{-1}$). For seasonal comparison of SOA formation rate caused by OOM condensation, the characteristic accumulation time (AccTime), defined as SOA divided by OOM condensation flux, is calculated as an indicator (see details in Sec. S3). As shown in Fig. 11 (B), a characteristic time of 24 hours is enough to explain the observed SOA concentration by OOM condensation in winter, and it is reduced to 7 hours, 10 hours, and 7 hours for spring, summer, and autumn, respectively. It should be noted that this should not be interpreted as the entire SOA being formed via OOM condensation during this characteristic time, but rather that OOM condensation is efficient and can have a significant contribution to SOA formation. A recent study (Nie et al., 2022) suggested that OOM condensation can account for about 40% of the SOA formation in wintertime Beijing. Our analysis on seasonal variation indicates that the condensation of OOMs could have a larger contribution to SOA formation in seasons other than winter.

For OOMs from different sources, aromatic OOMs contributes the most during the year, varying from 46 % to 62 %, followed by aliphatic OOMs (14-32 %). In comparison, the two biogenic ones, MT OOMs (8-12 %) and IP OOMs (3-5 %), have smaller contribution in all the four seasons. This indicates that the formation of SOA through condensation in urban Beijing is dominated by anthropogenic sources, which is in line with the previously reported SOA composition (Le Breton et al., 2018;Mehra et al., 2021). Overall, our results suggest that in order to control the formation of SOA, the emission of anthropogenic VOCs, especially aromatics, should be restricted with a high priority.

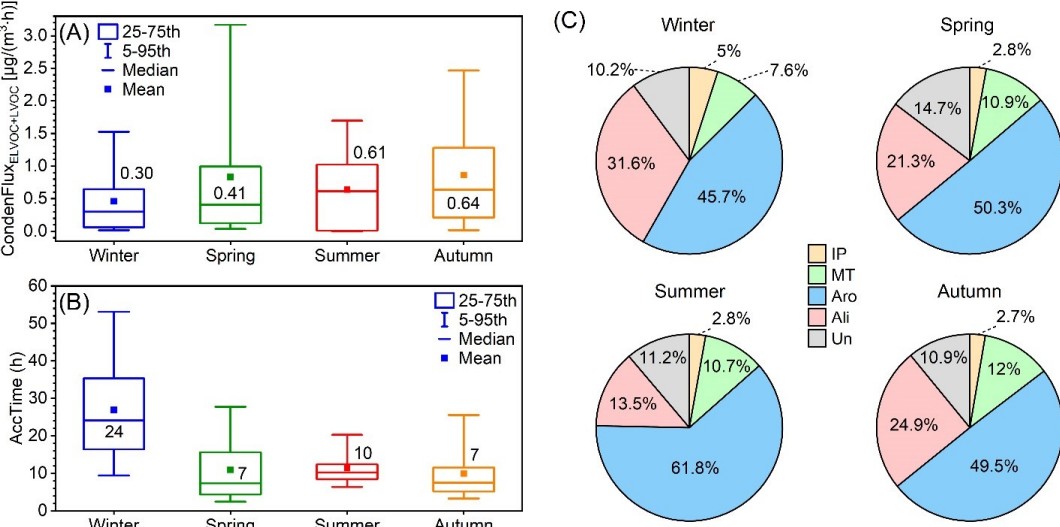


**Figure 11.** (A) Condensation flux of OOMs calculated by the particle dynamic model by Trostl et al. (Tröstl et al., 2016) in
four seasons**.** (B) Characteristic accumulation time of SOA (AccTime), calculated as SOA divided by OOM condensation flux,
in four season. This parameter is used as indicator for the relative accumulation rate of SOA caused by OOM condensation in
different seasons. The values in each box of (A) and (B) are the median values of corresponding parameters. (C) Estimated
condensation flux contribution of four source-classified OOMs in four seasons. The abbreviations "IP" "MT" "Aro" "Ali" and
"Un" stand for IP OOMs, MT OOMs, aromatic OOMs, aliphatic OOMs and undistinguished OOMs respectively.
**4. Summary and Conclusions**
A long-term measurement of OOMs was conducted in urban Beijing. Total OOM concentration in Beijing shows
a clear dependence on UVB and temperature, suggesting the importance of photo-oxidation and temperature on
OOM formation. In comparison to other atmospheric sites, total OOM concentration in Beijing ($2.3 \times 10^7$ - $1.6 \times 10^8$
$cm^{-3}$) is generally comparable to urban and suburban areas, and is clearly lower than those measured in forested
areas. In the case of composition, most OOMs have 5 to 10 carbon atoms, 3 to 7 effective oxygen atoms, 0 to 2
nitrogen atoms and 0 to 6 DBE values. The seasonal variation of average effective oxygen atom follows the same
trend as the overall atmospheric oxidation capacity, being the highest in summer and the lowest in winter and
autumn. While for nitrogen and DBE distribution, there are not too much difference among seasons disregarding
isoprene OOMs, indicating that the dominant formation pathways of OOMs stay the same during the year.
With a revised workflow, we further separate OOMs into isoprene, monoterpene, aromatic, and aliphatic OOMs.
For relative abundance, aromatic (29-41 %) and aliphatic OOMs (26-41 %) are major contributors throughout the
year, suggesting that OOMs in urban atmospheric environment are controlled by anthropogenic activities. In
addition, isoprene OOMs play an important role in summer and their fraction reaches to 33 %, indicating that
biogenic sources are also large contributors to total OOMs in warmer seasons. The concentration of isoprene OOMs



$(0.2\text{-}5.3\times10^7\ cm^{-3})$ and monoterpene OOMs $(1.1\text{-}8.4\times10^6\ cm^{-3})$ are smaller than those in forest areas, and they
exhibit higher nitrogen and lower oxygen content compared with other cleaner sites. One recent study (Nie et al.,
2022) reported that the composition of wintertime OOMs among four Chinese megacities, including Beijing, were
similar. Our study further demonstrates that the composition of summertime OOMs between Beijing and Nanjing
also have strong resemblance. Consequently, the seasonal characteristics of Beijing OOMs in this study could be
representative of OOMs in other Chinese metropolises.
In terms of volatility, monoterpene OOMs are the most condensable, isoprene OOMs are the most volatile, and
aromatic OOMs are more condensable than aliphatic ones. Based on the volatility and concentration characteristics
of the four source-classified OOMs, an aerosol growth model was utilized to calculate their contribution to SOA
growth. Results show that the condensation flux of total OOMs $(0.30\text{-}0.64\ \mu g\cdot m^{-3}\cdot h^{-1})$ are high enough to produce
a considerable amount of SOA within a day, and that aromatic (46-62 %) and aliphatic (14-32 %) OOMs are found
to be dominant contributors regardless of seasons. This suggests that the formation of SOA in urban cities are likely
driven by OOMs from anthropogenic sources, and highlights the importance of reducing anthropogenic emissions,
especially aromatics, for pollution mitigation.

**Data and materials availability:** Data and materials are available upon contacting the first author and
corresponding author.

**Author contributions:** CY and YG designed the study and wrote the manuscript. YG, YL, FZ, Ying Zhang, Ying
Zhou, CL, XF, ZL, ZF, Yusheng Zhang, PZ and LT conducted the measurement and collected the data. CY, WN,
ZW, DH, XQ, YL, YG, PZ and LT built the workflow and contributed to the aerosol dynamic model. JJ and VMK
modified the manuscript. And all co-authors have read and commented on the manuscript.

**Competing interests:** The authors declare no competing interest.

**Acknowledgements:** Heikki Junninen is acknowledged for providing the tofTool package used for processing
LTOF-CIMS data.

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
