# Peer review of "Seasonal Variation of Oxygenated Organic Molecules in Urban Beijing and their Contribution to Secondary Organic Aerosol"

_Atmospheric Chemistry and Physics, 2022_

## Referee Comment (RC1)

The manuscript "Seasonal variation of oxygenated organic molecules in urban Beijing and their contribution to secondary organic aerosol" addresses the seasonality of oxygenated organic molecules (OOM) in urban Beijing measured by a nitrate-CIMS over one year. And their potential contributions to secondary organic aerosol (SOA) via vapor condensation were estimated using an aerosol dynamic model. OOM concentrations were found to be highest in summer and generally followed the trend of radiation and temperature. OOM were classified into IP-OOM, MT-OOM, aromatic and aliphatic OOM. IP-OOM concentration vary strongly among seasons with lowest concentration in winter. Isoprene-OOM accounted for a large fraction in summer while in other seasons aromatic and aliphatic-OOM were the two major components. MS Profile of MT-OOM, aromatic and alphatic OOM were similar among different seasons. Condensation of aromatic- and aliphatic-OOM contributed most to SOA throughout the year.

Observations OOM in urban environment are important to elucidate the SOA formation and particle growth but are scare as most field observations of OOM were conducted in forested regions. Especially long time observation regarding seasonality of OOM is lacking. This study investigated one-year OOM composition and concentrations systematically, which provides very valuable information of OOM seasonality and potential contribution to SOA. The manuscript is well written and fits well the scope of ACP. I recommend its publication after addressing the following comments, which mainly work to clarify some important details and improve the readability and discussion of the manuscript.

**Specific comments**

1. L60, I suggest explicitly mentioning the definition of OOM.
2. In this study only mz 200-400 was analyzed. Why other mz such as higher mz were not included? What about their contributions? Are they too low to be important? It would be worth clarifying and noting the mz ranges in the conclusion.
3. In the method part (L173, Fig. 1), it seems that the authors used a IP-OOM list to differentiate whether a compound belongs to IP-OOM. It may be helpful to include the list and describe what experiments/observations the list is based on.
4. Also in this part and Fig. 3, have the authors considered the possible contribution of isoprene to C10-OOM via accretion reactions and contribution of monoterpene to C5-OOM via fragmentation?
5. Fig. 3, considering nitrate-CIMS does not detect OOM with low O number, are the OOM with nOeff<=4 mostly organic nitrates?
6. Fig. 5, it is not completely clear how the spectral profiles are obtained.
7. L262-264, Fig. 5, and Fig.7, analyzing day and night MS spectra separately and plotting the diurnal variation for IP-OOM and MT-OOM may be helpful to estimate the relative contribution of nitrate OOM from $NO_x$ termination and $NO_3$ radical.
8. L403-405, why CHO accounts for so low fraction compared to other OOM classes?
9. Fig. 11, how the volatility was calculated? As volatility highly depends on the method to derive it, it would be helpful to clarify and maybe add a short discussion on the influence of the methods adopted.
10. L493, "the dominant formation pathways of OOMs stay the same during the year", it might be easier to follow to add "…each OOM stay constant during the year", otherwise it would sound that all OOM are formed in the same pathways.

**Technical comments:**

1. L173, in Fig. 1, "nO=10" or "nC=10?

2. Fig. 3, the order of the legend is not consistent with the order of appearance. It might be easier to follow by keeping it consistent. And there is no legend for light blue.

3. L443, a space is missing before "94 %".

---

## Author Comment (AC1)

**Response to Reviewers**

**Referee #1**

The manuscript "Seasonal variation of oxygenated organic molecules in urban Beijing and their contribution to secondary organic aerosol" addresses the seasonality of oxygenated organic molecules (OOM) in urban Beijing measured by a nitrate-CIMS over one year. And their potential contributions to secondary organic aerosol (SOA) via vapor condensation were estimated using an aerosol dynamic model. OOM concentrations were found to be highest in summer and generally followed the trend of radiation and temperature. OOM were classified into IP-OOM, MT-OOM, aromatic and aliphatic OOM. IP-OOM concentration vary strongly among seasons with lowest concentration in winter. Isoprene-OOM accounted for a large fraction in summer while in other seasons aromatic and aliphatic-OOM were the two major components. MS Profile of MT-OOM, aromatic and aliphatic OOM were similar among different seasons. Condensation of aromatic- and aliphatic-OOM contributed most to SOA throughout the year. Observation OOMs in urban environment are important to elucidate the SOA formation and particle growth but are scare as most field observations of OOMs were conducted in forested regions. Especially long time observation regarding seasonality of OOM is lacking. This study investigated one-year OOM compositions and concentrations systematically, which provides very valuable information of OOM seasonality and potential contribution to SOA. The manuscript is well written and fits well with the scope of ACP. I recommend its publication after addressing the following comments, which mainly work to clarify some important details and improve the readability and discussion of the manuscript.

We thank the reviewer for the constructive comments and suggestions. And as suggested, we added more details and modified corresponding descriptions to make our analysis easier to follow. The point-to-point response to the comments is given below. And the comments, our replies, and the corresponding changes in the revised manuscript and supplementary information are marked in black, blue, and green texts, respectively.

**Specific comments**

1. L60, I suggest explicitly mentioning the definition of OOM.

Response: Thanks a lot and this is really necessary.

The terminology OOMs comes from the term HOMs (Highly Oxygenated Organic Molecules), which is defined based on three criteria (Bianchi et al., 2019): a. HOMs are formed via autoxidation involving peroxy radicals, b. HOMs are formed in the gas phase under atmospherically relevant conditions, and c. HOMs typically contain six or more oxygen atoms. However, when looking into the oxygenated organic molecules measured at urban Beijing, the criteria of a. and c. are not always met. For example, in the oxidation processes, precursor VOCs may undergo multi-generation oxidation (Zaytsev et al., 2019;Garmash et al., 2020) and the autoxidation is sometimes suppressed by high NOx level. This leads to the fact that not all organic molecules measured by the nitrate CIMS contain six or more oxygen atoms (see Fig. S7). This is the reason why we don't use the term HOMs.

OOMs is a more general terminology that refers to the gas phase organic compounds which are formed via oxidation process under atmospheric conditions. Thus, the guidelines for the classification of OOMs should be:

a. OOMs are the gas-phase oxygen-containing organic molecules detected under atmospherically relevant conditions.

b. OOMs can be formed via oxidations in the gas phase, heterogeneous reactions on aerosol surface, or evaporated processes from aerosols.

The definition of OOMs has been explicitly explained in a recent work by Nie et al. (Nie et al., 2022). Therefore, we added this paper as a reference of this sentence and explain the definition in more details in Sect. S2 (Line 22 - 35, Page 2).

"oxygenated organic molecules (OOMs, see the definition in Sect. S2 the work of Nie et al. (Nie et al., 2022)) have been acknowledged as an important contributor" (Line 59 – 60, Page 3)

2. In this study only m/z 200-400 was analyzed. Why other m/z, such as higher m/z were not included? What about their contributions? Are they too low to be important? It would be worth clarifying and noting the m/z range in the conclusion.

Response: We didn't report OOMs with higher m/z primarily because peaks in this range are very weak. Please note that those weak signals are not due to the transmission of the instrument, which remain roughly 94 % for m/z 400-680 range relative to that in the m/z 200-400 range. Rather, the low signal at higher m/z suggests that compounds of these masses are in very low concentrations. According to previous studies, the peaks with m/z larger than 400 are usually  $C_{15}$ - $C_{20}$  compounds. They can either be the products from the dimeric reactions of monoterpenes (Ehn et al., 2012;Jokinen et al., 2014;Yan et al., 2016), or the oxidations of larger VOCs, such as sesquiterpene and diterpenes (Richters et al., 2016;Luo et al., 2022). However, in our studied urban environment, the concentrations of sesquiterpene and diterpenes are very low, and the formation of dimeric compounds is greatly suppressed by high-level NOx. Therefore, the low signal at m/z > 400 can be well expected.

Meanwhile, we do observe some peaks at m/z > 400 range, but they are likely fluorine-containing contaminants and irrelevant to ambient processes. For example, the signal at m/z = 427 is mainly contributed by  $(C_3F_7COO)_2H^$ and the isotope of  $C_6H_{13}COOH \cdot NO_3^-$ , and remaining OOM signals are quite low (Fig. R1). The weak OOM signals together with the prominent contaminant signals make accurate peak identification difficult. Therefore, we decided not to focus too much on the higher m/z range, which nevertheless, has little influence on the conclusion of this study.

Figure R1. Fitted peaks at m/z = 427 Th of averaged winter spectrum (5th Jan. to 14th Feb., 2019).

For OOMs with m/z lower than 200 Th, their real molar weight after subtracting reagent ions should be low. Thus, they are generally more volatile and are not important to the condensational growth of SOA. One of the aim of this study is to investigate the SOA growth caused by OOM condensation, and this is the main reason why "light OOMs" are not included.

To clarify this consideration, we added the mass range of OOMs in the revised conclusion part as follows:

"A long-term measurement of OOMs based on nitrate CIMS was conducted in urban Beijing. OOMs in the mass range of 200-400 Th were systematically investigated." (Line 494 – 495, Page 21)

**3.** In the method part (L173, Fig. 1), it seems that the authors used a IP-OOM list to differentiate whether a compound belongs to IP-OOM. It may be helpful to include the list and describe what experiments/observations the list is based on.

Response: Thanks a lot for your suggestion. This isoprene OOM list is taken from the work of Xu et al. (Xu et al., 2021), where multifunctional products of isoprene oxidation were observed in both Nanjing and Shanghai. In this study, based on the molecular composition of precursor isoprene as well as current knowledge of atmospheric isoprene oxidation, the authors were able to constrain the double bond equivalence, numbers of hydrogen and number of effective oxygen of isoprene oxidation products. And according to those constrains, isoprene OOM molecules were selected.

We also added the isoprene OOM list in the revised supplement as Table S6.

Table S6. Peak list of isoprene OOMs in this study based on the work of Xu et al. (Xu et al., 2021)

| Molecular Formula                                                      | Exact Mass |  |  |  |  |  |
|-------------------------------------------------------------------------------|------------|--|--|--|--|--|
| СНО                                                                           |            |  |  |  |  |  |
| $C_4H_6O_4(NO_3)$                                                             | 180.0150   |  |  |  |  |  |
| C 5 H 8 O 4 (NO 3 - )  | 194.0306   |  |  |  |  |  |
| $C_5H_8O_5(NO_3)$                                                             | 210.0255   |  |  |  |  |  |
| $C_4H_6O_6(NO_3)$                                                             | 212.0048   |  |  |  |  |  |
| $C_5H_{10}O_5(NO_3)$                                                          | 212.0412   |  |  |  |  |  |
| $C_4H_8O_6(NO_3)$                                                             | 214.0205   |  |  |  |  |  |
| $C_5H_{12}O_5(NO_3)$                                                          | 214.0568   |  |  |  |  |  |
| $C_{5}H_{8}O_{6}(NO_{3})$                                                     | 226.0205   |  |  |  |  |  |
| $C_5H_{10}O_6(NO_3)$                                                          | 228.0361   |  |  |  |  |  |
| $C_4H_8O_7(NO_3)$                                                             | 230.0154   |  |  |  |  |  |
| $C_5H_{12}O_6(NO_3)$                                                          | 230.0518   |  |  |  |  |  |
| $C_{5}H_{8}O_{7}(NO_{3})$                                                     | 242.0154   |  |  |  |  |  |
| CHON                                                                          |            |  |  |  |  |  |
| C 5 H 9 O 4 N(NO 3 - ) | 209.0415   |  |  |  |  |  |
| $C_4H_7O_5N(NO_3)$                                                            | 211.0208   |  |  |  |  |  |
| $C_5H_9O_5N(NO_3)$                                                            | 225.0364   |  |  |  |  |  |
| $C_4H_7O_6N(NO_3)$                                                            | 227.0157   |  |  |  |  |  |
| $C_5H_9O_6N(NO_3)$                                                            | 241.0314   |  |  |  |  |  |
| $C_4H_7O_7N(NO_3)$                                                            | 243.0106   |  |  |  |  |  |
| $C_5H_{11}O_6N(NO_3)$                                                         | 243.0470   |  |  |  |  |  |
| $C_5H_9O_7N(NO_3)$                                                            | 257.0263   |  |  |  |  |  |
| $C_{5}H_{11}O_{7}N(NO_{3})$                                                   | 259.0419   |  |  |  |  |  |
| $C_5H_9O_8N(NO_3)$                                                            | 273.0212   |  |  |  |  |  |
| $C_4H_7O_9N(NO_3)$                                                            | 275.0004   |  |  |  |  |  |
| $C_5H_{11}O_8N(NO_3)$                                                         | 275.0368   |  |  |  |  |  |
| $C_5H_{11}O_9N(NO_3)$                                                         | 291.0317   |  |  |  |  |  |
| $C_5H_9O_{10}N(NO_3)$                                                         | 305.0110   |  |  |  |  |  |
| CHON 2,3                                                           |            |  |  |  |  |  |

| $C_5H_{10}O_6N_2(NO_3-)$                 | 256.0423 |
|------------------------------------------|----------|
| $C_4H_8O_7N_2$ (NO 3 -)       | 258.0215 |
| $C_5H_{10}O_7N_2(NO_3-)$                 | 272.0372 |
| $C_5H_{10}O_8N_2(NO_3-)$                 | 288.0321 |
| $C_5H_{10}O_9N_2(NO_3-)$                 | 304.0270 |
| $C_4H_8O_{10}N_2(NO_3-)$                 | 306.0063 |
| $C_5H_{10}O_{10}N_2$ (NO 3 -) | 320.0219 |
| $C_5H_9O_{10}N_3(NO_3-)$                 | 333.0172 |

4. Also in this part and Fig. 3, have the authors considered the possible contribution of isoprene to C10-OOM via accretion reactions and contribution of monoterpene to C5-OOM via fragmentation?

Response: We didn't incorporate the C5 OOMs formed from monoterpene (MT) fragmentation, nor the C10 OOMs formed from isoprene (IP) accretion in the workflow. This can result in some uncertainties, which however, are expected to be quite low.

As aforementioned, the list of IP oxidation products was taken from the work of Xu et al. (Xu et al., 2021), where all the IP OOMs are either C4 or C5 compounds. It has shown in chamber studies that via RO2 cross reactions, isoprene oxidation can lead to the formation of C10 dimeric products (Bernhammer et al., 2018;Wu et al., 2021), yet such RO2 cross reaction is severely suppressed by the high-level NOx (up to 9.5 - 38.3 ppb) in Beijing, limiting the formation of C10 isoprene products. Therefore, the C10 OOMs generated from isoprene are neglected in the workflow.

In previous experiments conducted in the CLOUD chamber, monoterpene oxidation by OH radical together with  $O_3$  in the presence of a few ppb  $NO_x$  have been investigated by Yan et al. (Yan et al., 2020). We revisited the data of one 4.5 ppb  $NO_x$  experiment and found that  $C_{4,5}$  fragmented peaks only accounted for ~ 4.5 % of total products (Fig. R2). In addition, the DBE values of most  $C_{4,5}$  fragments are larger than 2, whereas the highest DBE values of IP OOMs in this study is only 2 (Fig. 3). Thus, these fragments almost have no interference with the classification of IP OOMs. Furthermore, the total concentration of MT OOMs in this study is also much lower than IP OOMs. Therefore, a negligible fraction of IP OOMs could raise from fragmented MT OOMs by misclassification.

**Figure R2.** (A) Carbon distribution of all monoterpene oxidation products, and (B) DBE distribution of  $C_{4,5}$  monoterpene oxidation products formed by OH radical oxidation together with ozonolyis in previous experiment (Yan et al., 2020). The experiment was conducted under 278 K and the NOx level was 4.5 ppb.

5. Fig. 3, considering nitrate-CIMS does not detect OOM with low O number, are the OOM with  $nO_{eff} \le 4$  mostly organic nitrates?

Response: Yes. Most of  $nO_{eff} \le 4$  OOMs are organic nitrates.

The nitrogen number distributions of OOMs with  $nO_{eff} \le 4$  in four seasons are shown in Fig. R3 (A). It can be found that most OOM molecules contain one or more nitrogen atoms, and they are most likely organic nitrates. Besides, around 11 - 17 % of  $nO_{eff} \le 4$  OOMs contain no nitrogen atom, and most  $nO_{eff} \le 4$  and nN = 0 OOMs possess 4 oxygen atoms (Fig. R3 (B)).

**Figure R3.** (A) Distribution of nitrogen number (nN) for OOMs with number of effective oxygen no larger than 4 ( $nO_{eff} \le 4$ ). (B) Distribution of effective oxygen number ( $nO_{eff}$ ) for OOMs with  $nO_{eff} \le 4$  and nN = 0. The light blue, green, red and orange bars denote winter, spring, summer and autumn, respectively.

6. Fig. 5, it is not completely clear how the spectral profiles are obtained.

Response: In each subplot of Fig. 5, Fig. 7 and Fig. 9, the fraction of each compound is calculated via dividing its concertation by the corresponding total concentration of isoprene OOMs, monoterpene OOMs, aromatic OOMs or aliphatic OOMs.

To clarify this consideration, the captions of Fig. 5, Fig. 7 and Fig. 9 have been modified as follows:

"The fraction of each compound is calculated as the ratio of its concentration to the total concentration of IP OOMs." (Line 344 – 345, Page 13 for Fig. 5)

"The fraction of each compound is calculated as the ratio of its concentration to the total concentration of MT OOMs." (Line 372 - 373, Page 15 for Fig. 7)

"The fraction of each compound is calculated as the ratio of its concentration to the total concentration of aromatic OOMs ((A) to (D)) or aliphatic OOMs ((E) to (H))." (Line 432 - 433, Page 18 for Fig. 9)

7. L262-264, Fig. 5, and Fig.7, analyzing day and night MS spectra separately and plotting the diurnal variation for IP-OOM and MT-OOM may be helpful to estimate the relative contribution of nitrate OOM from  $NO_x$  termination and  $NO_3$  radical.

Response: Thank you very much for your suggestions.

The high resolution mass spectra of measured OOMs during day and night are plotted in Fig. R4. It can be found that although the concentrations of OOMs during the night are typically lower than during the day, the overall distributions and patterns of the mass spectra are very similar. Besides, there is almost no peak that is unique for day or night. Therefore, the it is not easy to determine the contribution of NO3 radical and NOx termination to nitrate OOM formation based on mass spectra.

---

## Author Comment (AC2)

**Response to Reviewers**

**Referee #2**

This manuscript presents the results of a field study of oxygenated organic molecules (OOMs) and their contribution to secondary organic aerosol (SOA) in urban Beijing. The measurements of OOMs were conducted over the four seasons of the year 2019 using a nitrate-CIMS. The measured OOMs mainly contained 5-10 carbon atoms, 3-7 effective oxygen atoms and 0-2 nitrogen atoms and had 0-6 DBE values. The OOM concentration exhibited an obvious seasonal variation, ranging from ~1 ppt in winter to ~7 ppt in summer. Such seasonality was thought to be mainly driven by the seasonal variation of the intensity of photochemistry. According to the DBE value and the number of carbon, oxygen and nitrogen atoms in the molecules, ~1000 OOMs were classified into four groups, that is, isoprene, monoterpene, aliphatic and aromatic OOMs. Among them, aromatic (29-41%) and aliphatic (26-41%) OOMs were found to be the major contributors to OOMs in all seasons. The vapor condensation flux calculations further showed that these two classes of OOMs had largest contributions (46-62% and 14-32%, respectively) to SOA in urban Beijing throughout the year. Up to date, field measurements of OOMs in urban areas are rare. This study provides valuable data on the concentration, chemical composition and seasonal variation of OOMs, as well as their potential contributions to SOA in polluted urban areas. Overall, the measurements and data analysis in this study are well performed, the results are appropriately discussed, and the manuscript is nicely written. I would recommend the publication of this manuscript in ACP after the following comments are fully addressed.

We thank the reviewer for the constructive comments and suggestions. We have carefully revised our manuscript and supplement accordingly. The point-to-point response to the comments is given below. And the comments, our replies, and the corresponding changes in the manuscript and supplementary information are in black, blue, and green, respectively.

1. Line 63-65: It is described here that API-TOF provided the first direct measurements of OOMs. That is true for semi-volatile and low-volatility OOMs. However, OOMs literally mean organic compounds with oxygenated functional groups and also include the family of oxygenated volatile organic compounds, which had been measured, e.g., by GC-MS and PTR-MS before API-ToF or ToF-CIMS have been developed. The authors should provide a clear definition or specify the range of OOMs discussed in this study.

Response: Thank you very much for your suggestion. Our description is not that rigorous and should be modified.

Here the OOMs mainly refer to the more oxygenated ones measured by Chemical Ionization-APi-ToF mass spectrometers, such as nitrate CIMS, iodide CIMS, and bromide CIMS. Therefore, we modified this sentence in the revised manuscript as follows:

"... the first direct measurement of highly oxygenated organic molecules (HOMs), a subgroup of OOMs with the most oxygen content, ..." (Line 64 - 65, Page 3)

For the terminology OOMs, it is a more general one that refers to the gas phase organic compounds which are formed via oxidation process under atmospheric conditions. We didn't use the term HOMs (Highly Oxygenated Organic Molecules) proposed by Bianchi (Bianchi et al., 2019) because a. not all OOMs are formed through autoxidation, and precursor VOCs may undergo multi-generation oxidation (Zaytsev et al., 2019;Garmash et al., 2020), and b. not all OOMs contain six or more oxygen atoms (see Fig. S7). Thus, the guidelines for the classification of OOMs in this study should be:

a. OOMs are the gas-phase oxygen-containing organic molecules detected under atmospherically relevant conditions.

b. OOMs can be formed via oxidations in the gas phase, heterogeneous reactions on aerosol surface, or evaporated processes from aerosols.

The definition of OOMs has been explicitly explained in a recent work by Nie et al. (Nie et al., 2022). Therefore, we added this paper as a reference of this sentence and explain the definition in more details in Sect. S2 (Line 22 - 35, Page 2).

"oxygenated organic molecules (OOMs, see the definition in Sect. S2 the work of Nie et al. (Nie et al., 2022)) have been acknowledged as an important contributor" (Line 59 - 60, Page 3)

2. Line 119-121: Since OOMs can be detected either as a nitrate ion cluster (i.e., [M+NO3-] or [M+HNO3•NO3-]) or as a deprotonated ion by nitrate-CIMS, additional information as to how the product ions [CHON+NO3-] vs. [CHO+ HNO3•NO3-] and [CHON-] vs. [CHO•NO3-] were differentiated in this study should be provided in the manuscript.

Response: Thanks a lot. This is indeed a piece of important information that needs to be provided.

The acidity of OOM molecules is not very strong so that they hardly exist in de-protonated ions. In Ehn et al., by using isotopically labeled nitric acid ( $H^{15}NO_3$ ) as the reagent, it was found that OOMs mainly existed as adducts with  $NO_3^-$ , and to a lesser extent with  $HNO_3NO_3^-$  (Ehn et al., 2014). This is because the binding energy of OOMs with  $NO_3^-$  is smaller than with  $HNO_3NO_3^-$  (Hyttinen et al., 2015).

Besides, the evidence that most OOMs cluster with  $NO_3^-$  could also be found from the time variations of corresponding OOMs. As shown in Fig. S1, the time variations of  $C_7H_{10}O_5NO_3^-$  and  $C_7H_{11}O_8NNO_3^-$  are not consistent, indicating that they are different OOMs charged both by  $NO_3^-$ , instead of the same OOM charged by  $NO_3^-$  and  $HNO_3NO_3^-$ , respectively. This is also the same for other OOMs, such as  $C_8H_{12}O_5NO_3^-$  and  $C_8H_{13}O_8NNO_3^-$ .